# Factors associated with admission to the intensive care unit and mortality in patients with COVID-19, Colombia

Jorge Enrique Machado-Alba[1]*, Luis Fernando Valladales-Restrepo[1,2], Manuel Enrique Machado-Duque[1,2], Andrés Gaviria-Mendoza[1,2], Nicolás Sánchez-Ramírez[1], Andrés Felipe Usma-Valencia[1], Esteban Rodríguez-Martínez[3], Eliana Rengifo-Franco[3], Víctor Hugo Forero-Supelano[4], Diego Mauricio Gómez-Ramírez[5], Alejandra Sabogal-Ortiz[5]

1 Grupo de Investigación en Farmacoepidemiología y Farmacovigilancia, Universidad Tecnológica de Pereira-Audifarma S.A, Pereira, Risaralda, Colombia, 2 Grupo de Investigación Biomedicina, Facultad de Medicina, Fundación Universitaria Autónoma de las Américas, Pereira, Colombia, 3 Semillero de Investigación en Farmacología Geriátrica, Facultad de Medicina, Fundación Universitaria Autónoma de las Américas, Pereira, Risaralda, Colombia, 4 Fundación Universitaria Juan N Corpas, Bogotá, D.C., Colombia, 5 Grupo Ospedale, Grupo Operador Clínico Hospitalario por Outsourcing S.A.S - G-Ocho S.A.S, Área de Salud, Cali, Valle del Cauca, Colombia

* machado@utp.edu.co

## Abstract

### Introduction

Coronavirus disease 2019 (COVID-19) has affected millions of people worldwide, and several sociodemographic variables, comorbidities and care variables have been associated with complications and mortality.

### Objective

To identify the factors associated with admission to intensive care units (ICUs) and mortality in patients with COVID-19 from 4 clinics in Colombia.

### Methods

This was a follow-up study of a cohort of patients diagnosed with COVID-19 between March and August 2020. Sociodemographic, clinical (Charlson comorbidity index and NEWS 2 score) and pharmacological variables were identified. Multivariate analyses were performed to identify variables associated with the risk of admission to the ICU and death (p<0.05).

### Results

A total of 780 patients were analyzed, with a median age of 57.0 years; 61.2% were male. On admission, 54.9% were classified as severely ill, 65.3% were diagnosed with acute respiratory distress syndrome, 32.4% were admitted to the ICU, and 26.0% died. The factors associated with a greater likelihood of ICU admission were severe pneumonia (OR: 9.86; 95%CI:5.99–16.23), each 1-point increase in the NEWS 2 score (OR:1.09; 95%

**Data Availability Statement:** Data and material available at protocolos.io (dx.doi.org/10.17504/protocols.io.bud5ns86).

**Funding:** The author(s) received no specific funding for this work.

CI:1.002–1.19), history of ischemic heart disease (OR:3.24; 95%CI:1.16–9.00), and chronic obstructive pulmonary disease (OR:2.07; 95%CI:1.09–3.90). The risk of dying increased in those older than 65 years (OR:3.08; 95%CI:1.66–5.71), in patients with acute renal failure (OR:6.96; 95%CI:4.41–11.78), admitted to the ICU (OR:6.31; 95%CI:3.63–10.95), and for each 1-point increase in the Charlson comorbidity index (OR:1.16; 95%CI:1.002–1.35).

## Conclusions

Factors related to increasing the probability of requiring ICU care or dying in patients with COVID-19 were identified, facilitating the development of anticipatory intervention measures that favor comprehensive care and improve patient prognosis.

## Introduction

In Wuhan (China), at the end of 2019, a series of cases of pneumonia caused by a new corona-virus were reported [1]. The pathogen was named SARS-CoV-2 (severe acute respiratory syndrome coronavirus 2) by the International Committee on Taxonomy of Viruses (ICTV), and the pneumonia it produced was called coronavirus disease-2019 (COVID-19) by the World Health Organization (WHO) [2]. On January 30, 2020, COVID-19 was declared an epidemic of international concern [3]. This infection has affected tens of millions of people worldwide, leading to millions of deaths [4]. In Colombia, according to the National Institute of Health and the Ministry of Health, confirmed cases exceed 2 million, with more than 56000 deaths (2.6%), mostly males (63.7%) and those over 60 years (78.4%) [5].

This has led to an unprecedented burden on health systems worldwide, including increased hospital admissions, high demand for intensive care unit (ICU) beds, advanced respiratory support, renal replacement therapy, and other life support interventions [6]. The impact of the COVID-19 pandemic on health systems varies by country, depending on the balance between the supply and demand of services, which has been associated with the ability to expand the number of hospital beds, particularly in the ICU, and public health policies to contain the pandemic [6, 7].

Although the majority of people with SARS-CoV-2 have mild or uncomplicated disease, 14% develop serious disease requiring oxygen therapy, and approximately 5% require treatment in an ICU; of these, most require mechanical ventilation [8]. Among the prognostic factors described for the development of critical illness and mortality, the most important are advanced age; the presence of certain comorbidities, such as arterial hypertension, diabetes, chronic obstructive pulmonary disease, cardiovascular disease, and obesity; abnormalities in some paraclinical tests; and the availability of medications [2, 9]. These factors should be recognized by attending physicians to identify critical patients early, to allocate resources effectively and to adapt management plans to improve patient prognosis [2].

Considering the information from international studies conducted to date, it is important to have epidemiological data at local and national scales because those data may differ from findings in North America, Europe and Asia. Although COVID-19 is a global pandemic, the burden of disease has not been the same in different countries [7], and in this sense, the aim herein was to identify the factors associated with ICU admission and with mortality in patients with COVID-19 in a Colombian population.

## Materials and methods

This was an observational study of the factors associated with ICU admission and with mortality in patients with COVID-19, who were identified from a report of positive cases confirmed by RT-PCR (reverse transcription polymerase chain reaction) testing in 4 tertiary care clinics affiliated with the *Grupo Ospedale Network*, located in the cities of Bogotá, Cali, Pereira and Popayán. All subjects of any age, sex and city of residence treated for COVID-19 between March 6 and August 31, 2020 were selected. Each patient was followed until death or hospital discharge. Those with incomplete medical records or incomplete follow-up by teleconsultation and those diagnosed by screening were excluded.

Based on the information obtained, a database was designed to collect the following groups of variables:

1. Sociodemographic: sex, age, city of origin, occupation and place of care (city/department).

2. Clinical:

   a. Physiological variables: body mass index (BMI), mean blood pressure, heart rate, respiratory rate, oxygen saturation, state of consciousness at the time of the emergency room care, physical examination (i.e: crackles, rhonchi, etc);

   b. Comorbidities: hypertension, diabetes mellitus, dyslipidemia, hypothyroidism, ischemic heart disease, heart failure, chronic obstructive pulmonary disease, asthma, solid tumors or hematological malignancies, human immunodeficiency virus infection, rheumatologic diseases (rheumatoid arthritis, systemic lupus erythematosus, vasculitis, and others), chronic kidney disease, stroke, obesity, and smoking, among others. The age-adjusted Charlson comorbidity index was calculated;

   c. Symptoms and signs: documented in the clinical record at hospital admission (i.e: Cough, fever, dyspnea, etc);

   d. Diagnostic intervention: laboratory tests (blood count, creatinine, urea nitrogen, total bilirubin, direct bilirubin, transaminases, lactate dehydrogenase, C-reactive protein, ferritin, D-dimer, troponin I and prothrombin time) and diagnostic imaging (initial chest X-ray and computerized axial tomography (CT-scan) of the chest) at the time of care;

   e. Service: emergency room, hospitalization in general wards and ICU; and

   f. Hospital stay and ICU stay (in days): date of admission and discharge (from the hospital and from the ICU) or date of death.

3. Therapeutic intervention (pharmacological):

   a. Therapy prescribed for COVID-19: antimalarials (chloroquine and hydroxychloroquine), azithromycin, lopinavir-ritonavir, tocilizumab, colchicine, ivermectin and convalescent plasma;

   b. Other medications: systemic corticosteroids (oral and parenteral), systemic antibiotics, vasopressors and inotropes (norepinephrine, vasopressin, and dopamine, among others), parenteral anticoagulants, sedatives (benzodiazepines, dexmedetomidine, and others), muscle relaxants, analgesics (non-opioids), antihypertensives and diuretics, normoglycemic agents, antiulcer drugs, benzodiazepines, bronchodilators and inhaled corticosteroids, and antipsychotics, among others;

 c. Use of supplemental oxygen: low-flow devices (nasal cannula/simple face mask and non-rebreathing face mask) and high-flow devices (venturi system and noninvasive and invasive mechanical ventilation). For patients who required mechanical ventilation, the total duration in days of orotracheal intubation (date of intubation and date of definitive extubation) was determined; and

 d. Position of the patient: prone.

4. Severity of COVID-19: classified according to the Colombian Consensus of Care, Diagnosis and Management of SARS-CoV-2/COVID 19 infection in health care facilities [8]. The CURB-65 and NEWS 2 scores were calculated for all patients, and the APACHE-II score was calculated for critically ill patients.

5. Complications: acidosis, acute heart injury, acute kidney injury, acute respiratory distress syndrome, arrhythmia, coagulopathy, complications of mechanical ventilation, cytokine-related syndrome, delirium, heart failure, kidney replacement therapy (dialysis), respiratory failure, secondary infection, sepsis, shock, spontaneous pneumothorax, thromboembolism.

6. Primary outcomes: admission to the ICU and death.

The protocol was approved by the Bioethics Committee of the Universidad Tecnológica de Pereira in the category of "risk-free research" (approval code: 03–080620). The principles of confidentiality of information established by the Declaration of Helsinki were respected. All data were fully anonymized before accessed them and the Bioethics Committee waived the requirement for informed consent.

The data were analyzed with the statistical package SPSS Statistics, version 26.0 for Windows (IBM, USA). A descriptive analysis was performed; frequencies and proportions are reported for the qualitative variables, and measures of central tendency and dispersion are reported for the quantitative variables, depending on their parametric behavior established by the Kolmogorov-Smirnov test. Quantitative variables were compared using Student's t-test or the Mann-Whitney U test and $X^2$ or Fisher's exact test for categorical variables. Exploratory binary logistic regression models were developed using ICU admission or death as the dependent variable. Covariates included age, sex and those variables that were significantly associated the dependent variables in the bivariate analysis. The level of statistical significance was established as $p < 0.05$.

## Results

A total of 780 patients with a confirmed diagnosis of SARS-CoV-2 were identified; the patients were from 48 different cities and were treated at 4 clinics in the country. A total of 477 (61.2%) were male, and the median age was 57.0 years (interquartile range [IQR]: 45.0–68.0 years; range: 0–100 years). The distribution by age group can be seen in Table 1. A total of 4.9% (n = 38) of the patients had a health-related job. The 4.6% (n = 14/303) of women were pregnant.

The most common comorbidities were hypertension, diabetes mellitus, obesity and chronic obstructive pulmonary disease. The median age-adjusted Charlson comorbidity index was 2 points (IQR: 0–3 points), and 33.9% (n = 264) had a score of 3 or more (see Table 1). The symptoms most reported by patients were cough, fever and dyspnea. At the time of admission, 46.0% had an oxygen saturation <90%, 38.1% had a heart rate of 100 or higher, and 25.7% (n = 194) had a respiratory rate of 24 or higher. Among the findings on physical examination, the presence of decreased breath sounds and wheezing was highlighted (see Table 1).

**Table 1. Sociodemographic, pharmacological, clinical and comorbidity variables among patients who survived and died, infected by SARS-CoV-2.**

| Characteristics | Total | | Survivors | | Deceased | | p |
|---|---|---|---|---|---|---|---|
| | n = 780 | % | n = 577 | % | n = 203 | % | |
| **Sociodemographic** | | | | | | | |
| Male | 477 | 61.2 | 350 | 60.7 | 127 | 62.6 | 0.632 |
| Female | 303 | 38.8 | 227 | 39.3 | 76 | 37.4 | |
| Age, median (IQR) | 57.0 (45.0–68.0) | | 53.0 (42.0–64.0) | | 67.0 (58.0–76.0) | | <0.001^ |
| <40 years | 126 | 16.2 | 122 | 21.1 | 4 | 2.0 | <0.001 |
| 40–64 years | 392 | 50.3 | 316 | 54.8 | 76 | 37.4 | <0.001 |
| 65–79 years | 196 | 25.1 | 108 | 18.7 | 88 | 43.3 | <0.001 |
| ≥80 years | 66 | 8.5 | 31 | 5.4 | 35 | 17.2 | <0.001 |
| **City of Attention** | | | | | | | |
| Bogotá | 306 | 39.2 | 247 | 42.8 | 59 | 29.1 | 0.001 |
| Cali | 302 | 38.7 | 190 | 32.9 | 112 | 55.2 | <0.001 |
| Pereira | 100 | 12.8 | 76 | 13.2 | 24 | 11.8 | 0.621 |
| Popayan | 72 | 9.2 | 64 | 11.1 | 8 | 3.9 | 0.002 |
| **Comorbidities** | | | | | | | |
| Charlson index, median (IQR) | 2 (0–3) | | 1 (0–2.5) | | 3 (2–5) | | <0.001^ |
| 0 points | 217 | 27.8 | 202 | 35.0 | 15 | 7.4 | <0.001 |
| 1–2 points | 299 | 38.3 | 231 | 40.0 | 68 | 33.5 | 0.099 |
| 3–4 points | 166 | 21.3 | 97 | 16.8 | 69 | 34.0 | <0.001 |
| ≥5 points | 98 | 12.6 | 47 | 8.1 | 51 | 25.1 | <0.001 |
| Arterial hypertension | 299 | 38.3 | 184 | 31.9 | 115 | 56.7 | <0.001 |
| Diabetes mellitus | 160 | 20.5 | 106 | 18.4 | 54 | 26.6 | 0.013 |
| Obesity | 132 | 16.9 | 92 | 15.9 | 40 | 19.7 | 0.219 |
| Chronic obstructive pulmonary disease | 75 | 9.6 | 36 | 6.2 | 39 | 19.2 | <0.001 |
| Hypothyroidism | 61 | 7.8 | 45 | 7.8 | 16 | 7.9 | 0.97 |
| Chronic kidney disease | 57 | 7.3 | 26 | 4.5 | 31 | 15.3 | <0.001 |
| Tobacco use | 53 | 6.8 | 37 | 6.4 | 16 | 7.9 | 0.474 |
| Dyslipidemia | 35 | 4.5 | 26 | 4.5 | 9 | 4.4 | 0.966 |
| Heart failure | 34 | 4.4 | 19 | 3.3 | 15 | 7.4 | 0.014 |
| Ischemic heart disease | 26 | 3.3 | 8 | 1.4 | 18 | 8.9 | <0.001 |
| Other comorbidities | 160 | 20.5 | 98 | 17.0 | 62 | 30.5 | <0.001 |
| **Medication history** | | | | | | | |
| Antihypertensives and diuretics | 249 | 31.9 | 163 | 28.2 | 86 | 42.4 | <0.001 |
| Antidiabetic agents | 117 | 15.0 | 80 | 13.9 | 37 | 18.2 | 0.134 |
| Analgesics | 63 | 8.1 | 56 | 9.7 | 7 | 3.4 | 0.005 |
| Lipid-lowering drugs | 57 | 7.3 | 39 | 6.8 | 18 | 8.9 | 0.321 |
| Thyroid hormone | 49 | 6.3 | 36 | 6.2 | 13 | 6.4 | 0.934 |
| **Symptoms** | | | | | | | |
| Cough | 570 | 73.1 | 435 | 75.4 | 135 | 66.5 | 0.014 |
| Fever / chills | 555 | 71.2 | 415 | 71.9 | 140 | 69.0 | 0.424 |
| Dyspnea | 525 | 67.3 | 363 | 62.9 | 162 | 79.8 | <0.001 |
| Fatigue | 324 | 41.5 | 243 | 42.1 | 81 | 39.9 | 0.582 |
| Myalgias / arthralgias | 226 | 29.0 | 185 | 32.1 | 41 | 20.2 | 0.001 |
| Headache | 158 | 20.3 | 133 | 23.1 | 25 | 12.3 | 0.001 |
| Odynophagia | 157 | 20.1 | 139 | 24.1 | 18 | 8.9 | <0.001 |
| Constitutional symptoms | 102 | 13.1 | 69 | 12.0 | 33 | 16.3 | 0.118 |
| Chest pain | 100 | 12.8 | 74 | 12.8 | 26 | 12.8 | 0.995 |

(*Continued*)

**Table 1.** (Continued)

| Characteristics | Total | | Survivors | | Deceased | | p |
|---|---|---|---|---|---|---|---|
| | **n = 780** | **%** | **n = 577** | **%** | **n = 203** | **%** | |
| Diarrhea | 89 | 11.4 | 71 | 12.3 | 18 | 8.9 | 0.185 |
| **Vital signs (on admission)** | | | | | | | |
| Mean arterial pressure (mmHg), median (IQR) | 93.3 (83.8–101.3) | | 93.3 (84.7–100.8) | | 93.7 (82.0–103.3) | | 0.805^ |
| <65 mmHg | 13 | 1.7 | 4 | 0.7 | 9 | 4.4 | 0.002* |
| Heart rate (beats/minute), median (IQR) | 91.5 (80.0–108) | | 90.0 (80.0–106.0) | | 95.0 (80.0–110.0) | | 0.012^ |
| ≥ 100 beats/minute | 289 | 38.1 | 197 | 35.5 | 92 | 45.3 | 0.014 |
| Temperature (˚C), median (IQR) | 36.5 (36.0–37.0) | | 36.5 (36.0–37.0) | | 36.5 (36.0–37.0) | | 0.430^ |
| > 38 ˚ C | 47 | 6.3 | 35 | 6.4 | 12 | 6.0 | 0.843 |
| Respiratory rate (breaths/minute), median (IQR) | 20.0 (18.0–24.0) | | 20 (18–22) | | 20 (19–25) | | <0.001^ |
| ≥ 24 breaths/minute | 194 | 25.7 | 121 | 21.9 | 73 | 36.0 | <0.001 |
| Oxygen saturation (%), median (IQR) | 90.0 (84.0–94.0) | | 91.0 (85.0–94.0) | | 87.0 (77.0–92.0) | | <0.001^ |
| <90% | 348 | 46.0 | 220 | 39.6 | 128 | 63.4 | <0.001 |
| **Physical examination (upon admission)** | | | | | | | |
| Body mass index (kg/m$^2$), median (IQR) | 27.1 (24.4–29.7) | | 27.3 (25.3–30.6) | | 26.7 (23.5–28.4) | | 0.014^ |
| ≥30.0 kg/m$^2$ | 71 | 24.3 | 47 | 28.5 | 24 | 18.9 | 0.058 |
| Decreased breath sounds | 169 | 21.7 | 113 | 19.6 | 56 | 27.6 | 0.017 |
| Crackles | 157 | 20.1 | 104 | 18.0 | 53 | 26.1 | 0.013 |
| Rhonchi | 89 | 11.4 | 58 | 10.1 | 31 | 15.3 | 0.044 |
| Intercostal retractions | 57 | 7.3 | 32 | 5.5 | 25 | 12.3 | 0.001 |
| Wheezing | 23 | 2.9 | 14 | 2.4 | 9 | 4.4 | 0.146 |

IQR: Interquartile range;

* Fisher's exact test;

^ Mann-Whitney U Test

Chest X-rays taken at admission showed abnormalities in 50.3% (n = 392) of patients, with infiltrates being the most frequent finding (n = 287; 36.8%); the predominant feature in CT scans was ground-glass opacity (n = 364; 46.7%). Table 2 describes the radiological and laboratory results.

For the CURB-65 criteria, the median score was 1 (IQR:0–1), with the majority of patients scoring between 0 and 1 point (n = 608; 77.9%), followed by 2 points (n = 125; 16.0%) and 3 or more points (n = 47; 6.0%). A total of 54.9% (n = 428) of patients had severe pneumonia on admission, 46.5% (n = 363) had a high-risk NEWS 2 score, and the median APACHE II score for 175 patients was 10 (IQR: 8–17). The median overall hospital stay was 7 days (IQR:4–12). The 10.6% (n = 83) of the patients only required care in the emergency room; 83.2% (n = 649) required care in a general ward, with a median stay of 6 days (IQR:3–9), and 32.4% (n = 253) were admitted to the ICU, with a median stay of 8 days (IQR:4–14). A total of 68.2% (n = 532) of all patients presented some type of complication, in particular acute respiratory distress syndrome (n = 509; 65.3%), and 26.0% (n = 203) died. Table 3 shows the complications suffered by patients who survived and those who died.

A total of 674 (86.4%) patients required supplemental oxygen, especially through low-flow devices (n = 662; 84.9%), in particular, nasal cannulas or simple face masks (n = 630; 80.8%) and non-rebreathing face masks (n = 313; 40.1%), while high-flow devices were used for 30.9% (n = 241) of all patients, in particular, invasive mechanical ventilation (n = 203; 26.0%), venturi devices (n = 76; 9.7%) and noninvasive mechanical ventilation (n = 21; 2.7%). A total of 26.4% (n = 206) of all patients were placed in the prone position. Of the patients who required

**Table 2. Laboratory and imaging studies at the time of initial care among patients who survived and died, infected by SARS-CoV-2.**

| Characteristics | Total | | Survivors | | Deceased | | p |
|---|---|---|---|---|---|---|---|
| | n = 780 | % | n = 577 | % | n = 203 | % | |
| **Laboratory studies, median (IQR)** | | | | | | | |
| **Blood count** | | | | | | | |
| Hemoglobin (g / dL) | 14.4 (13.1–15.5) | | 14.6 (13.4–15.7) | | 13.8 (12.2–14.8) | | <0.001^ |
| Hematocrit (%) | 42.6 (38.7–46.3) | | 43.2 (39.9–46.5) | | 40.6 (37.0–45.3) | | <0.001^ |
| Leukocytes (/ mm$^3$) | 8.840 (6.280–11.710) | | 8.155 (6.000–10.885) | | 9.980 (8.000–14.125) | | <0.001^ |
| Neutrophils (/ mm$^3$) | 6.870 (4.450–9.630) | | 6.235 (4.100–8.915) | | 8.650 (6.087–11.925) | | <0.001^ |
| Lymphocytes (/ mm$^3$) | 1.000 (730–1.400) | | 1.050 (792–1.500) | | 844 (600–1.240) | | <0.001^ |
| Platelets (mil / mm$^3$) | 248.0 (190.0–309.0) | | 254.0 (208.2–311.7) | | 225.0 (171.0–277.0) | | 0.001^ |
| **Renal function** | | | | | | | |
| Creatinine (mg / dL) | 0.9 (0.7–1.1) | | 0.9 (0.7–1.1) | | 1.1 (0.8–1.6) | | <0.001^ |
| Urea nitrogen (mg / dL) | 16.2 (12.5–23.7) | | 15.3 (12.1–21.7) | | 24.4 (17.0–36.9) | | <0.001^ |
| **Liver function** | | | | | | | |
| Total bilirubin (mg / dL) | 0.57 (0.39–0.92) | | 0.52 (0.37–0.81) | | 0.65 (0.44–0.90) | | 0.008^ |
| Direct bilirubin (mg / dL) | 0.31 (0.20–0.49) | | 0.26 (0.18–0.40) | | 0.40 (0.25–0.56) | | <0.001^ |
| Alanine aminotransferase (U / L) | 43.4 (26.1–61.0) | | 38.8 (26.0–59.4) | | 42.9 (27.0–68.0) | | 0.377^ |
| Aspartate aminotransferase (U / L) | 43.0 (29.6–61.0) | | 42.8 (28.2–61.0) | | 54.9 (33.0–85.0) | | 0.011^ |
| Lactic dehydrogenase (U / L) | 367.0 (284.0–489.0) | | 359.0 (282.7–454.7) | | 458.0 (347.0–611.0) | | <0.001^ |
| **Others** | | | | | | | |
| C-reactive protein (mg / L) | 130.5 (63.6–206.4) | | 116.9 (49.4–186.1) | | 166.5 (105.7–257.5) | | <0.001^ |
| Ferritin (ng / mL) | 1010.5 (451.0–1906.7) | | 899.0 (408.0–1657.4) | | 1340.0 (555.3–2000.0) | | 0.004^ |
| D-dimer (μg / mL) | 340.0 (20.0–648.0) | | 299.5 (1.763–558.5) | | 497.0 (283.0–1141.0) | | <0.001^ |
| Troponin I (ng / mL) | 0.008 (0.004–0.019) | | 0.007 (0.004–0.011) | | 0.024 (0.008–0.055) | | <0.001^ |
| Prothrombin time (sec) | 13.9 (10.6–15.3) | | 12.9 (10.2–15.2) | | 14.1 (10.9–15.4) | | 0.023^ |
| **Imaging studies** | | | | | | | |
| **Chest x-ray** | | | | | | | |
| Abnormal findings | 392 | 50.3 | 290 | 50.3 | 102 | 50.2 | 0.997 |
| Infiltrate | 287 | 36.8 | 211 | 36.6 | 76 | 37.4 | 0.825 |
| Consolidation | 94 | 12.1 | 67 | 11.6 | 27 | 13.3 | 0.525 |
| Groud-glass opacity | 89 | 11.4 | 68 | 11.8 | 21 | 10.3 | 0.579 |
| Pleural effusion | 26 | 3.3 | 18 | 3.1 | 8 | 3.9 | 0.575 |
| Atelectasis | 23 | 2.9 | 18 | 3.1 | 5 | 2.5 | 0.634 |
| Air bronchogram | 10 | 1.3 | 7 | 1.2 | 3 | 1.5 | 0.726* |
| **Chest Computed Tomography** | | | | | | | |
| Abnormal findings | 395 | 50.6 | 276 | 47.8 | 119 | 58.6 | 0.008 |
| Groud-glass opacity | 364 | 46.7 | 250 | 43.3 | 114 | 56.2 | 0.002 |
| Consolidation | 118 | 15.1 | 91 | 15.8 | 27 | 13.3 | 0.398 |
| Air bronchogram | 81 | 10.4 | 38 | 6.6 | 43 | 21.2 | <0.001 |
| Atelectasis | 45 | 5.8 | 31 | 5.4 | 14 | 6.9 | 0.423 |
| Lymphadenopathy | 38 | 4.9 | 29 | 5.0 | 9 | 4.4 | 0.736 |
| Bronchiectasis | 35 | 4.5 | 21 | 3.6 | 14 | 6.9 | 0.054 |
| Interlobular septal thickening | 28 | 3.6 | 12 | 2.1 | 16 | 7.9 | <0.001 |
| Pleural thickening | 27 | 3.5 | 19 | 3.3 | 8 | 3.9 | 0.664 |

IQR: Interquartile range;

* Fisher's exact test;

^ Mann-Whitney U Test

**Table 3. Complications among patients who survived and died, infected with SARS-CoV-2.**

| Characteristics | Total | | Survivors | | Deceased | | p |
|---|---|---|---|---|---|---|---|
| | n = 780 | % | n = 577 | % | n = 203 | % | |
| **Complications** | 532 | 68.2 | 329 | 57.0 | 203 | 100.0 | <0.001 |
| Acute respiratory distress syndrome | 509 | 65.3 | 314 | 54.4 | 195 | 96.1 | <0.001 |
| Admission to ICU | 253 | 32.4 | 94 | 16.3 | 159 | 78.3 | <0.001 |
| Respiratory failure | 184 | 23.6 | 29 | 5.0 | 155 | 76.4 | <0.001 |
| Acute kidney injury | 171 | 21.9 | 46 | 8.0 | 125 | 61.6 | <0.001 |
| Acidosis | 103 | 13.2 | 26 | 4.5 | 77 | 37.9 | <0.001 |
| Shock | 89 | 11.4 | 10 | 1.7 | 79 | 38.9 | <0.001 |
| Secondary infection | 81 | 10.4 | 27 | 4.7 | 54 | 26.6 | <0.001 |
| Sepsis | 75 | 9.6 | 12 | 2.1 | 63 | 31.0 | <0.001 |
| Kidney replacement therapy (dialysis) | 64 | 8.2 | 10 | 1.7 | 54 | 26.6 | <0.001 |
| Complications of mechanical ventilation | 42 | 5.4 | 7 | 1.2 | 35 | 17.2 | <0.001 |
| Arrhythmia | 37 | 4.7 | 4 | 0.7 | 33 | 16.3 | <0.001 |
| Acute heart injury | 21 | 2.7 | 6 | 1.0 | 15 | 7.4 | <0.001 |
| Coagulopathy | 21 | 2.7 | 9 | 1.6 | 12 | 5.9 | 0.001 |
| Heart failure | 19 | 2.4 | 3 | 0.5 | 16 | 7.9 | <0.001* |
| Delirium | 15 | 1.9 | 5 | 0.9 | 10 | 4.9 | 0.001* |
| Thromboembolism | 5 | 0.6 | 2 | 0.3 | 3 | 1.5 | 0.114* |
| Spontaneous pneumothorax | 4 | 0.5 | 3 | 0.5 | 1 | 0.5 | 1.000* |
| Cytokine-related syndrome | 2 | 0.3 | 1 | 0.2 | 1 | 0.5 | 0.453* |

ICU: Intensive care unit.

* Fisher's exact test.

invasive mechanical ventilation, the median duration of intubation was 8 days (IQR:4–15; range: 0–66 days). The most commonly used drugs in this group of patients were antimicrobials (n = 633; 81.2%), anticoagulants (n = 614; 78.7%), and systemic corticosteroids (n = 462; 59.2%). Table 4 outlines the pharmacological treatment received by patients.

Patients in the city of Cali were significantly older and had higher NEWS 2 scores at admission, higher rates of severe pneumonia and a higher requirement for invasive mechanical ventilation than patients in other cities (see S1 Table).

## Multivariate analysis

The binary logistic regression found that in the city of Cali, ischemic heart disease, chronic obstructive pulmonary disease, severe pneumonia, and each 1-point increase in NEWS 2 score increased the probability of being admitted to an ICU. No variables were found that reduced this risk (Table 5). Being 65 or older, each 1-point increase in the Charlson comorbidity index, presenting severe pneumonia, requiring ICU care and presenting complications such as acute respiratory distress syndrome and acute kidney failure were associated with a greater probability of death. There were also no variables that reduced this risk (Table 6).

## Discussion

The present study identified factors related to increasing the probability of ICU admission or death in a group of patients with confirmed SARS-CoV-2 treated in 4 cities in Colombia. The identification of these risk factors will allow intervention measures to be proposed and thus contribute to improving the prognosis of these patients [10].

**Table 4. Pharmacological management received during medical care among patients who survived and died, infected by SARS-CoV-2.**

| Characteristics | Total | | Survivors | | Deceased | | p |
|---|---|---|---|---|---|---|---|
| | n = 780 | % | n = 577 | % | n = 203 | % | |
| Antibiotics (without azithromycin) | 633 | 81.2 | 436 | 75.6 | 197 | 97.0 | <0.001 |
| Ampicillin sulbactam | 502 | 64.4 | 360 | 62.4 | 142 | 70.0 | 0.053 |
| Clarithromycin | 278 | 35.6 | 205 | 35.5 | 73 | 36.0 | 0.912 |
| Cefepime | 124 | 15.9 | 39 | 6.8 | 85 | 41.9 | <0.001 |
| Vancomycin | 96 | 12.3 | 27 | 4.7 | 69 | 34.0 | <0.001 |
| Ceftriaxone | 90 | 11.5 | 61 | 10.6 | 29 | 14.3 | 0.154 |
| Meropenem | 82 | 10.5 | 19 | 3.3 | 63 | 31.0 | <0.001 |
| Anticoagulants | 614 | 78.7 | 424 | 73.5 | 190 | 93.6 | <0.001 |
| Enoxaparin | 577 | 74.0 | 411 | 71.2 | 166 | 81.8 | 0.003 |
| Unfractionated heparin | 45 | 5.8 | 17 | 2.9 | 28 | 13.8 | <0.001 |
| Dalteparin | 7 | 0.9 | 1 | 0.2 | 6 | 3.0 | 0.002* |
| Antiulcer drugs | 566 | 72.6 | 392 | 67.9 | 174 | 85.7 | <0.001 |
| Analgesics | 501 | 64.2 | 378 | 65.5 | 123 | 60.6 | 0.208 |
| Non-opioid analgesics | 473 | 60.6 | 363 | 62.9 | 110 | 54.2 | 0.029 |
| Opioid analgesics | 96 | 12.3 | 53 | 9.2 | 43 | 21.2 | <0.001 |
| Systemic corticosteroids | 462 | 59.2 | 321 | 55.6 | 141 | 69.5 | 0.001 |
| Dexamethasone | 429 | 55.0 | 308 | 53.4 | 121 | 59.6 | 0.125 |
| Hydrocortisone | 46 | 5.9 | 10 | 1.7 | 36 | 17.7 | <0.001 |
| Methylprednisolone | 22 | 2.8 | 11 | 1.9 | 11 | 5.4 | 0.009 |
| Prednisolone or prednisone | 14 | 1.8 | 6 | 1.0 | 8 | 3.9 | 0.013* |
| Proposed COVID-19 therapy | | | | | | | |
| Azithromycin | 257 | 32.9 | 169 | 29.3 | 88 | 43.3 | <0.001 |
| Ivermectin | 118 | 15.1 | 95 | 16.5 | 23 | 11.3 | 0.079 |
| Colchicine | 81 | 10.4 | 57 | 9.9 | 24 | 11.8 | 0.435 |
| Antimalarials | 33 | 4.2 | 19 | 3.3 | 14 | 6.9 | 0.028 |
| Hydroxychloroquine | 19 | 2.4 | 12 | 2.1 | 7 | 3.4 | 0.293* |
| Chloroquine | 15 | 1.9 | 8 | 1.4 | 7 | 3.4 | 0.077* |
| Lopinavir / ritonavir | 14 | 1.8 | 5 | 0.9 | 9 | 4.4 | 0.003* |
| Plasma | 5 | 0.6 | 0 | 0.0 | 5 | 2.5 | 0.001* |
| Tocilizumab | 1 | 0.1 | 0 | 0.0 | 1 | 0.5 | 0.260 |
| Inhaled bronchodilators and corticosteroids | 319 | 40.9 | 249 | 43.2 | 70 | 34.5 | 0.031 |
| Antihypertensives and diuretics | 280 | 35.9 | 175 | 30.3 | 105 | 51.7 | <0.001 |
| Benzodiazepines (without midazolam) | 216 | 27.7 | 51 | 8.8 | 165 | 81.3 | <0.001 |
| Sedatives | 210 | 26.9 | 41 | 7.1 | 169 | 83.3 | <0.001 |
| Midazolam | 208 | 26.7 | 41 | 7.1 | 167 | 82.3 | <0.001 |
| Fentanyl | 197 | 25.3 | 38 | 6.6 | 159 | 78.3 | <0.001 |
| Dexmedetomidine | 104 | 13.3 | 31 | 5.4 | 73 | 36.0 | <0.001 |
| Ketamine | 104 | 13.3 | 24 | 4.2 | 80 | 39.4 | <0.001 |
| Antidiabetic agents | 194 | 24.9 | 105 | 18.2 | 89 | 43.8 | <0.001 |
| Muscle relaxants | 186 | 23.8 | 38 | 6.6 | 148 | 72.9 | <0.001 |
| Vasopressors and inotropics | 180 | 23.1 | 29 | 5.0 | 151 | 74.4 | <0.001 |
| Antipsychotics | 102 | 13.1 | 37 | 6.4 | 65 | 32.0 | <0.001 |

* Fisher's exact test.

**Table 5. Binary logistic regression of variables associated with the probability of admission to the intensive care unit in patients with a diagnosis of SARS-CoV-2.**

| Characteristics | p | OR | 95% CI | |
| --- | --- | --- | --- | --- |
| | | | Lower | Upper |
| Male sex | 0.256 | 1.252 | 0.85 | 1.844 |
| Age ≥65 years | 0.39 | 1.21 | 0.784 | 1.867 |
| Cali (city of residence) | <0.001 | 3.153 | 2.149 | 4.625 |
| Health related profession | 0.536 | 1.47 | 0.435 | 4.971 |
| Obesity | 0.083 | 1.521 | 0.947 | 2.445 |
| Ischemic heart disease | 0.024 | 3.243 | 1.167 | 9.009 |
| Diabetes mellitus | 0.053 | 1.564 | 0.995 | 2.457 |
| Chronic kidney disease | 0.805 | 0.918 | 0.468 | 1.803 |
| Chronic obstructive pulmonary disease | 0.026 | 2.066 | 1.093 | 3.904 |
| Arterial hypertension | 0.338 | 1.229 | 0.806 | 1.875 |
| Non-opioid analgesics | 0.149 | 0.509 | 0.204 | 1.273 |
| Severe pneumonia | <0.001 | 9.865 | 5.995 | 16.232 |
| NEWS2 score | 0.044 | 1.087 | 1.002 | 1.179 |

NEWS2: National Early Warning Score 2. OR: Odds ratio. 95% CI: 95% confidence interval.

The median age of patients with COVID-19 was similar to that found in other studies (56.0–72.0 years) [11–17], with a predominance of males, as also identified in most studies (51.9–62.0%) [11–13, 15–18], except in a cohort of patients in the USA where a higher proportion of females (55.9%) was described [14]. Regarding these sociodemographic variables, some studies have found that males have a higher risk of complications [14, 19, 20] and death [14, 18, 20]; those findings were not identified in this report, but our results are consistent with those described in other studies [15, 21, 22]. It was observed that as age increased, patients had a higher probability of dying, consistent with a large number of international publications [14, 17, 21–24] and local studies [11, 25], probably due to a higher disease burden [10].

**Table 6. Binary logistic regression of variables associated with the probability of dying in patients diagnosed with SARS-CoV-2.**

| Characteristics | p | OR | 95% CI | |
| --- | --- | --- | --- | --- |
| | | | Lower | Upper |
| Male sex | 0.187 | 0.718 | 0.439 | 1.175 |
| Age ≥65 years | <0.001 | 3.087 | 1.667 | 5.719 |
| Cali (city of residence) | 0.394 | 1.312 | 0.702 | 2.452 |
| Charlson Comorbidity Index | 0.046 | 1.164 | 1.002 | 1.351 |
| Severe pneumonia | 0.008 | 2.463 | 1.263 | 4.801 |
| NEWS2 score | 0.559 | 1.031 | 0.931 | 1.142 |
| Admission to ICU | <0.001 | 6.309 | 3.634 | 10.954 |
| Antimalarials | 0.075 | 0.407 | 0.151 | 1.096 |
| Azithromycin | 0.794 | 1.084 | 0.592 | 1.985 |
| Corticosteroids | 0.960 | 0.986 | 0.570 | 1.705 |
| Systemic antibiotics | 0.416 | 0.625 | 0.201 | 1.941 |
| Acute kidney injury | <0.001 | 6.966 | 4.116 | 11.788 |
| Acute respiratory distress syndrome | 0.007 | 3.448 | 1.408 | 8.445 |

NEWS2: National Early Warning Score 2. ICU: intensive care units. OR: Odds ratio. 95% CI: 95% confidence interval.

The most frequent comorbidities in this cohort of patients were hypertension and diabetes mellitus, a finding that is consistent with those in other reports [11, 12, 14–17, 20, 22, 23]. These pathologies have been associated with a greater probability of presenting complications and severe forms of the disease [10, 26]; however, this was not the case in this study and in some previously published works [14, 23, 27]. However, ischemic heart disease and chronic obstructive pulmonary disease did increase the risk of complications requiring ICU admission, consistent with what was found by other authors [27–29]. Likewise, the probability of dying increased 16% for each 1-point increase in the Charlson comorbidity index, a finding similar to that documented in a group of patients in Spain (OR:1.23; 95%CI:1.15–1.32) [28] and in a cohort of patients in the USA; in those cohorts, mortality increased from 40 to 93% depending the scoring method [19], making it a useful tool to identify patients with a higher risk of mortality and therefore those who require closer clinical monitoring [30].

The clinical manifestations most described in the literature have been cough, fever and dyspnea [11, 14, 15, 17, 22, 23], as found in this analysis. With respect to the NEWS 2 score, which involves different clinical parameters, this report showed that a higher score was associated with a greater probability of requiring ICU care, a result that is consistent with a study conducted in Colombia (Bogotá), where the score was associated with a greater risk of disease severity (for each 1-point increase, (HR:1.15; 95%CI:1.03–1.28) [11]. Different studies have identified that this scale applied at hospital admission is a good predictor of severe disease, ICU admission and mortality in patients with COVID-19 [31, 32].

In addition, some laboratory test results have been associated with an increased risk of mortality [2, 9, 11, 15, 17, 19, 23, 33], such as elevated levels of creatinine [19, 33], C-reactive protein [15, 33], lactate dehydrogenase [11], transaminases [19], and D-dimer [17] or low levels of monocytes [15], lymphocytes [19], and albumin [19], among others [2, 9]. In this report, significant differences were found in paraclinical testing between patients who survived and those who died, similar to that reported in the literature [11, 17, 23] However, paraclinical testing was not included in the multivariate models because the data were not available for all patients. The radiological finding most frequently found was ground-glass opacification, consistent with what has been reported in the literature [11, 17, 27]; this finding is one of the typical characteristics on CT chest scans in the early phases of COVID-19 [34].

In this cohort of patients, more than half were initially classified as having serious disease, similar to that found in China (63.0%) [17], and higher than that previously published in Colombia (31.7%) [11]; serious disease is related to complications and mortality [11]. Among the complications, almost two-thirds of the patients presented with acute respiratory distress syndrome, which has also been found frequently in other studies but at various proportions (24.1%-90.0%) [7, 11, 14, 17, 23, 24]. In addition, 23.6% of the patients in this study progressed to respiratory failure, consistent with what was reported in a systematic review and meta-analysis (16.2%; 95%CI:0.4–43.3%) [9]. This clinical condition was also associated with an increased risk of death, consistent with what was found in Italy [20] but not in Spain [23]. Another complication that also presented a significant association with mortality was acute renal failure, a risk that was previously documented by Ferrando et al. in Spain (OR:2.46; 95% CI:1.62–3.74) [23].

While patients treated in the city of Cali were more likely to be admitted to the ICU, they were less likely to die. The latter is in line with what was found in a study conducted with data from the National Public Health Surveillance System (SIVIGILA) in several cities of Colombia; patients in Valle del Cauca had a lower risk of mortality than did patients in the rest of the country (relative risk:0.81; 95%CI:0.73–0.90; p<0.001) [18]. With respect to the higher risk of admission to the ICU in Cali, this is probably because in Cali, the patients were older and had

a higher NEWS 2 score and there were more severe cases on admission, factors that led to many of these patients requiring ICU care [11, 14, 19, 20, 31, 32].

Regarding the management received by these patients, just over a quarter required invasive mechanical ventilation, a finding that is consistent with other reports (12.2%-23.0%) [12, 13, 16, 17], and almost one-third required ICU care, a proportion that was similar to that found in other countries (26.0%-39.7%) [13, 14, 17]. In cohorts of hospitalized patients in Spain, Iran and Italy, most were managed with antimalarials [23, 33, 35] or azithromycin [23], but in this analysis, these drugs were used in less than one-third of patients; systemic corticosteroids were used similarly to what was published in other studies (60.9%-76.3%) [23, 35]. In this report, none of the therapies were associated with improving the prognosis of patients, a result that is consistent with those reported in several studies [21, 28]. The current evidence (at the time of writing this manuscript) indicates that antimalarials, azithromycin and ivermectin do not reduce complications or mortality in patients with COVID-19 [8, 36, 37]; however, a randomized clinical trial (RECOVERY) showed that the use of dexamethasone reduced the risk of death by 36% in patients who received invasive mechanical ventilation and by 18% among patients who required supplemental oxygen [38]. Notably, this drug may be associated with a higher frequency of bacterial infections and electrolyte disorders [39].

Observational studies have certain limitations that should be taken into account when interpreting the results. In this study, because the information was only obtained from the data recorded from a group of patients from 4 tertiary care clinics located in different cities, the findings may not extrapolate to all types of health care institutions or to all regions of the country. In addition, for some variables, especially those related to clinical laboratory tests, information was not available for all patients; therefore, the inclusion of these types of variables in the multivariate analyses was limited.

## Conclusions

Based on these findings, it can be concluded that having some comorbidities, such as ischemic heart disease or chronic obstructive pulmonary disease, prior to the diagnosis of COVID-19, suffering from severe pneumonia, each 1-point increase in NEWS 2 score and being treated in the city of Cali increased the probability of being admitted to an ICU. Advanced age, especially >65 years, each 1-point increase in the Charlson comorbidity index, severe pneumonia, complications, such as acute respiratory distress syndrome or acute renal failure, and requiring ICU care increased the probability of death. No variables were identified that would reduce the risk of requiring ICU care or of dying. These results can be useful for clinicians who care for patients with COVID-19 because the recognition of these variables can be used to improve the quality of care.

## Supporting information

**S1 Table. Comparison of some sociodemographic and clinical variables among the cities of care of a group of patients infected by SARS-CoV-2, Colombia.**
(DOCX)

## Acknowledgments

### Declarations

**Author responsibility**. The corresponding author confirm full access to all data in the study and final responsibility.

## Author Contributions

**Conceptualization:** Jorge Enrique Machado-Alba, Luis Fernando Valladales-Restrepo, Manuel Enrique Machado-Duque.

**Data curation:** Luis Fernando Valladales-Restrepo, Andrés Gaviria-Mendoza, Nicolás Sánchez-Ramírez, Andrés Felipe Usma-Valencia, Esteban Rodríguez-Martínez, Eliana Rengifo-Franco, Víctor Hugo Forero-Supelano.

**Formal analysis:** Luis Fernando Valladales-Restrepo, Manuel Enrique Machado-Duque, Nicolás Sánchez-Ramírez.

**Investigation:** Jorge Enrique Machado-Alba, Manuel Enrique Machado-Duque, Andrés Gaviria-Mendoza, Andrés Felipe Usma-Valencia, Esteban Rodríguez-Martínez, Eliana Rengifo-Franco, Diego Mauricio Gómez-Ramirez.

**Methodology:** Jorge Enrique Machado-Alba, Manuel Enrique Machado-Duque, Andrés Gaviria-Mendoza.

**Project administration:** Jorge Enrique Machado-Alba.

**Resources:** Víctor Hugo Forero-Supelano, Diego Mauricio Gómez-Ramirez, Alejandra Sabogal-Ortiz.

**Software:** Andrés Gaviria-Mendoza, Alejandra Sabogal-Ortiz.

**Supervision:** Jorge Enrique Machado-Alba, Diego Mauricio Gómez-Ramirez, Alejandra Sabogal-Ortiz.

**Validation:** Víctor Hugo Forero-Supelano, Alejandra Sabogal-Ortiz.

**Writing – original draft:** Luis Fernando Valladales-Restrepo.

**Writing – review & editing:** Jorge Enrique Machado-Alba.

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
