## [Decision Letter · Decision Letter 0]

13 Jul 2021

PONE-D-21-18267

Factors associated with admission to the intensive care unit and mortality in patients with COVID-19, Colombia

PLOS ONE

Dear Dr. Machado-Alba,

Thank you for submitting your manuscript to PLOS ONE. After careful consideration, we feel that it has merit but does not fully meet PLOS ONE’s publication criteria as it currently stands. Therefore, we invite you to submit a revised version of the manuscript that addresses the points raised during the review process.

We look forward to receiving your revised manuscript.

Kind regards,

Tai-Heng Chen, M.D.

Academic Editor

PLOS ONE

2. In your ethics statement in the manuscript and in the online submission form, please ensure that you have discussed whether all data/samples were fully anonymized before you accessed them and/or whether the IRB or ethics committee waived the requirement for informed consent. If patients provided informed written consent to have data/samples from their medical records used in research, please include this information.

3. In the ethics statement in the manuscript and in the online submission form, please provide additional information about the patient records/samples used in your retrospective study, including  the source of the medical records/samples analyzed in this work (e.g. hospital, institution or medical center name).

Reviewers' comments:

Reviewer's Responses to Questions

**Comments to the Author**

1. Is the manuscript technically sound, and do the data support the conclusions?

Reviewer #1: Yes

Reviewer #2: No

2. Has the statistical analysis been performed appropriately and rigorously? 

Reviewer #1: I Don't Know

Reviewer #2: No

3. Have the authors made all data underlying the findings in their manuscript fully available?

Reviewer #1: Yes

Reviewer #2: No

4. Is the manuscript presented in an intelligible fashion and written in standard English?

Reviewer #1: Yes

Reviewer #2: Yes

5. Review Comments to the Author

Reviewer #1: Authors have submitted a manuscript of significant importance. It is essential to present COVID-19 related data that originate outside developed countries since there is obvious and dramatic disparity in availability and quality if patient care between developed and developing countries. I recommend acceptance after minor but careful revision. Specific comments are listed below.

1. Reconsider keywords.

2. In Introduction: The first paragraph is redundant. Overall , Introduction should be shorter and more to the point.

3. A phrase „behavior of the infection“ should be reconsidered.

4. In Methods: A sentence „Subjects of any age, sex and city of residence were selected between March 6 and August 31, 2020.” should be rephrased to clearly state if all patients admitted to hospital during this time frame were screened for inclusion.

5. In Methods: Which CXR or CT scans were used? The ones on admission to hospital or the ones with the worst scores. COVID-19 is fast evolving and repeated CXRs and CT scans are a necessity.

6. Line 112: symptoms and signs should be listed in methods. In fact, list of all variables shown in Table 1 and 2 should be listed in methods. Those variables that are not shown in Results and used for statistical analysis should be omitted.

7. Line 118: „in general“ should be substituted for „from the hospital“.

8. Line 130/131: Were there any patients with tracheotomy?

9. Line 160/161: Sentence „among the females.....were pregnant“ should definitely undergo respectful rephrasing.

10. Line 199: „ days of intubation“ referes to duration of intubation or days between hospital admission and intubation?

11. Tables 1 to 4 show differences between survivors and non survivors. Somehow, we have „jumped“ from there to prediction of ICU admission. Steps before prediction modeling should be shown as well, process of selection of variables included in the prediction model should be clear. On the same note, it is not clear how selection of variables included in the mortality prediction was performed.

Reviewer #2: I would like to thank the editors of Plos One for giving me the opportunity to review the manuscript “factors associated with admission to the intensive care unit and mortality in patients with COVID-19 in Colombia”.

In this observational cohort study, the authors described the characteristics of 780 patients admitted to 4 clinics for COVID-19 in Colombia between march 6 and august 31, 2020 and tried to identify factor associated with death or ICU admission. The authors confirmed previous risk factors for poor outcomes described in much larger cohorts, including from south America. They also identified that, at the time of the study, patients from Cali had poorer outcomes.

This report appears now as completely anachronic since both viruses and treatments have profoundly changed. The study provides no new information that can be relevant for other caregivers.

Another major limitation of this study is that authors provided comparison between survivors and deceased patients (they report in the methods that they followed-up patient until death…) whereas the main result (as indicated in the title) was the factors associated with either ICU admission or death. In multivariate analysis, the found that coming from Cali, COPD and severe pneumoniae were associated with ICU admission or death. I really don’t understand how this finding may “greatly contribute to improving the prognosis of these patients” as concluded by the authors.

6. PLOS authors have the option to publish the peer review history of their article (what does this mean?). If published, this will include your full peer review and any attached files.

Reviewer #1: **Yes: **Suzana Bojic

Reviewer #2: No

---

## [Author Response · Author response to Decision Letter 0]

28 Jul 2021

Pereira, July 23 of 2021

Response to reviewers

Journal: PLOS ONE

Manuscript ID: PONE-D-21-18267

Title: Factors associated with admission to the intensive care unit and mortality in patients with COVID-19, Colombia

Thank you for the review. Here, we answer the reviewers’ comments point by point. 

Comments to the Author

1. Is the manuscript technically sound, and do the data support the conclusions?

Reviewer #1: Yes

Reviewer #2: No

2. Has the statistical analysis been performed appropriately and rigorously?

Reviewer #1: I Don't Know

Reviewer #2: No

3. Have the authors made all data underlying the findings in their manuscript fully available?

Reviewer #1: Yes

Reviewer #2: No

4. Is the manuscript presented in an intelligible fashion and written in standard English?

Reviewer #1: Yes

Reviewer #2: Yes

5. Review Comments to the Author

Reviewer #1: Authors have submitted a manuscript of significant importance. It is essential to present COVID-19 related data that originate outside developed countries since there is obvious and dramatic disparity in availability and quality if patient care between developed and developing countries. I recommend acceptance after minor but careful revision. Specific comments are listed below.

Response/ Thank you

1. Reconsider keywords.

Response/ we have adjusted the keywords

2. In Introduction: The first paragraph is redundant. Overall, Introduction should be shorter and more to the point.

Response/ the introduction has been shortened and some phrases have been simplified. 

3. A phrase „behavior of the infection“ should be reconsidered.

Response/ The phrase was simplified. 

4. In Methods: A sentence „Subjects of any age, sex and city of residence were selected between March 6 and August 31, 2020.” should be rephrased to clearly state if all patients admitted to hospital during this time frame were screened for inclusion.

Response/ The phrase has been adjusted. 

5. In Methods: Which CXR or CT scans were used? The ones on admission to hospital or the ones with the worst scores. COVID-19 is fast evolving and repeated CXRs and CT scans are a necessity.

Response/ We report these images on admission. We have now clarified this in the methods section. 

6. Line 112: symptoms and signs should be listed in methods. In fact, list of all variables shown in Table 1 and 2 should be listed in methods. Those variables that are not shown in Results and used for statistical analysis should be omitted.

Response/ The variables shown in tables / results are now also consistent with those in the methods section. 

7. Line 118: „in general“ should be substituted for „from the hospital“.

Response / Corrected

8. Line 130/131: Were there any patients with tracheotomy?

Response/ This information is not available, we did not searched this variable. 

9. Line 160/161: Sentence „among the females.....were pregnant“ should definitely undergo respectful rephrasing.

Response/ we rephrased the sentence.

10. Line 199: „ days of intubation“ referes to duration of intubation or days between hospital admission and intubation?

Response/ This refers to “duration”. The phrase has been adjusted accordingly.

11. Tables 1 to 4 show differences between survivors and non survivors. Somehow, we have „jumped“ from there to prediction of ICU admission. Steps before prediction modeling should be shown as well, process of selection of variables included in the prediction model should be clear. On the same note, it is not clear how selection of variables included in the mortality prediction was performed.

Response/ We included survival status in the tables in order to show the distribution of the variables among survivors and deceased. The step-by-step process for variable selection is not shown explicitly, but we have now changed the sentence in methods in this regard. We now explain that “Covariates included age, sex and those variables that were significantly associated the dependent variables in the bivariate analysis”. We have also indicated that this is an exploratory analysis, not a predictive one. 

Reviewer #2: I would like to thank the editors of Plos One for giving me the opportunity to review the manuscript “factors associated with admission to the intensive care unit and mortality in patients with COVID-19 in Colombia”.

In this observational cohort study, the authors described the characteristics of 780 patients admitted to 4 clinics for COVID-19 in Colombia between march 6 and august 31, 2020 and tried to identify factor associated with death or ICU admission. The authors confirmed previous risk factors for poor outcomes described in much larger cohorts, including from south America. They also identified that, at the time of the study, patients from Cali had poorer outcomes.

This report appears now as completely anachronic since both viruses and treatments have profoundly changed. The study provides no new information that can be relevant for other caregivers.

Response/ Thank you, but we believe this data are still of importance, especially in the colombian context.

Another major limitation of this study is that authors provided comparison between survivors and deceased patients (they report in the methods that they followed-up patient until death…) whereas the main result (as indicated in the title) was the factors associated with either ICU admission or death. 

Response/ We indeed reported both outcomes (i.e: table 5 and 6 show multivariate analysis considering each of these outcomes). We did not use a composite outcome of ICU admission + death. 

In multivariate analysis, the found that coming from Cali, COPD and severe pneumoniae were associated with ICU admission or death. I really don’t understand how this finding may “greatly contribute to improving the prognosis of these patients” as concluded by the authors.

Response/ The phrase has been adjusted. 

Thank you, we will be aware of new observations. 

The authors

---

## [Decision Letter · Decision Letter 1]

4 Nov 2021

Factors associated with admission to the intensive care unit and mortality in patients with COVID-19, Colombia

PONE-D-21-18267R1

Dear Dr. Machado-Alba,

We’re pleased to inform you that your manuscript has been judged scientifically suitable for publication and will be formally accepted for publication once it meets all outstanding technical requirements.

Kind regards,

Wenbin Tan

Academic Editor

PLOS ONE

Reviewers' comments:

Reviewer's Responses to Questions

Reviewer #3: All comments have been addressed

---

## [Editor Report · Acceptance letter]

9 Nov 2021

PONE-D-21-18267R1 

Factors associated with admission to the intensive care unit and mortality in patients with COVID-19, Colombia 

Dear Dr. Machado-Alba:

I'm pleased to inform you that your manuscript has been deemed suitable for publication in PLOS ONE. Congratulations! Your manuscript is now with our production department. 

Kind regards, 

on behalf of

Dr. Wenbin Tan 

Academic Editor

PLOS ONE